# Learning Frequency Domain Approximation for Binary Neural Networks

**Yixing Xu**[1], **Kai Han**[1], **Chang Xu**[2], **Yehui Tang**[1,3], **Chunjing Xu**[1], **Yunhe Wang**[*1]
[1]Huawei Noah's Ark Lab
[2]The University of Sydney   [3]Peking University
{yixing.xu, kai.han, tangyehui, xuchunjing, yunhe.wang}@huawei.com
c.xu@sydney.edu.au

## Abstract

Binary neural networks (BNNs) represent original full-precision weights and activations into 1-bit with sign function. Since the gradient of the conventional sign function is almost zero everywhere which cannot be used for back-propagation, several attempts have been proposed to alleviate the optimization difficulty by using approximate gradient. However, those approximations corrupt the main direction of factual gradient. To this end, we propose to estimate the gradient of sign function in the Fourier frequency domain using the combination of sine functions for training BNNs, namely frequency domain approximation (FDA). The proposed approach does not affect the low-frequency information of the original sign function which occupies most of the overall energy, and high-frequency coefficients will be ignored to avoid the huge computational overhead. In addition, we embed a noise adaptation module into the training phase to compensate the approximation error. The experiments on several benchmark datasets and neural architectures illustrate that the binary network learned using our method achieves the state-of-the-art accuracy. Code will be available at *https://gitee.com/mindspore/models/tree/master/research/cv/FDA-BNN*.

## 1   Introduction

The success of deep convolutional neural networks (CNNs) has been well demonstrated in several real-world applications, *e.g.*, image classification [24, 35, 59, 16], object detection [42], semantic segmentation [34], and low-level computer vision [46]. Massive parameters and huge computational complexity are usually required for achieving the desired high performance, which limits the application of these models to portable devices such as mobile phones and smart cameras.

In order to reduce the computational costs of deep neural networks, a number of works have been proposed to compress and accelerate the original cumbersome model into a portable one. For example, filter pruning methods [33, 60, 8, 58, 47, 48] aim to sort the filters based on their importance scores in which unimportant filters will be treated as redundancy and removed to derive lightweight architectures. Knowledge distillation methods [19, 36, 5, 52, 51] are explored to transfer the knowledge inside a teacher model to a student model. Tensor decomposition method [40] decomposes the original large weight tensor into several small ones to simplify the overall inference process. Model quantization methods [13, 22, 54, 32, 31] quantize the widely used 32-bit floating point weights and activations into low-bit ones to save the memory consumption and computational cost.

---

[*]Corresponding author.

35th Conference on Neural Information Processing Systems (NeurIPS 2021).

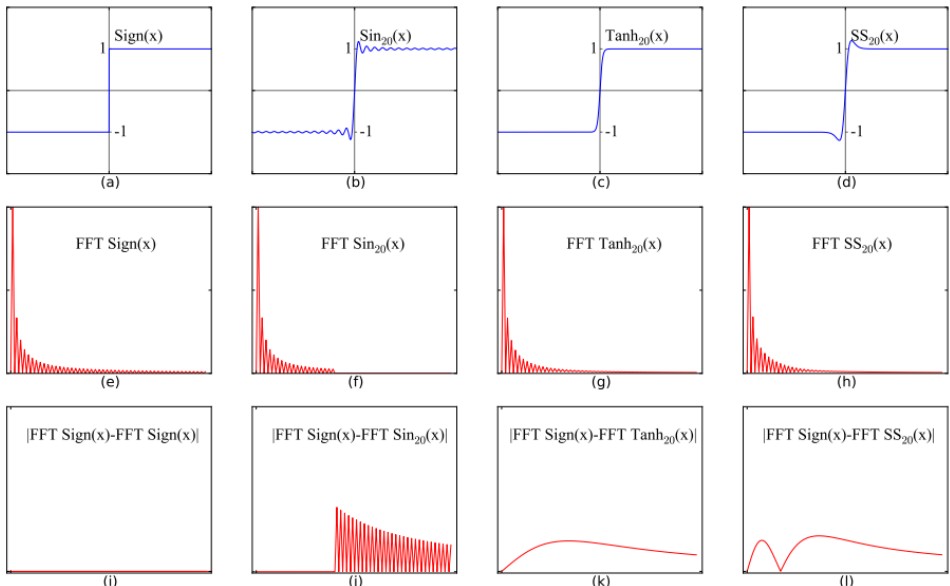

Figure 1: From top to bottom: original functions in spatial domain, corresponding functions in frequency domain and the difference between the current function and sign function in frequency domain. From left to right: sign function, combination of sine functions, tanh function in [10] and SignSwish function in [6] (short as SS).

Among the aforementioned algorithms, binary neural network (BNN) is an extreme case of model quantization method in which the weights and activations in the original neural network will be quantized to 1-bit (-1 and +1) values. Obviously, the memory usage for weights and activations in binary convolutional layers are $32\times$ lower than that in the full-precision network with the same model architecture. BNN was firstly proposed by Hubara *et.al.* [22] by utilizing straight-through estimator (STE) [1] for handling gradient problem of sign function. Then, XNOR-Net [41] introduced a channel-wise scaling factor for weights to reduce the quantization error. Dorefa-Net [57] used a single scaling factor for all the channels and achieved competitive performance. In addition, there are a number of methods for optimizing the original structure to improve the performance of learned BNN. ABCNet [26] used multiple parallel binary convolutional layers. Bi-Real Net [30] inserted skip-connections to enhance the feature representation ability of deep BNNs.

The optimization difficulty caused by the sign function in BNNs is a very important issue. In practice, the original full-precision weights and activations will be quantized into binary ones with sign function during the feed-forward procedure, but the gradient of sign function cannot be directly used for back-propagation. Therefore, most of the approaches mentioned above adopt the straight-through estimator (STE) [1] for gradient back-propagation to train BNNs. The inconsistency between the forward and backward pass caused by the STE method prevents BNN from achieving better performance. Additionally, several attempts were proposed to estimate the gradient of sign function for replacing STE. For example, DSQ [10] introduced a tanh-alike differentiable asymptotic function to estimate the forward and backward procedures of the conventional sign function. BNN+ [6] used a SignSwish activation function to modify the back-propagation of the original sign function and further introduced a regularization that encourages the weights around binary values. RBNN [25] proposed a training-aware approximation function to replace the sign function when computing the gradient. Although these methods have made tremendous effort for refining the optimization of binary weights and activations, the approximation function they used may corrupt the **main direction** of factual gradient for optimizing BNNs.

To better maintain the main direction of factual gradient in BNNs, we propose a frequency domain approximation approach by utilizing the Fourier Series (FS) to estimate the original sign function in the frequency domain. The FS estimation is a lossless representation of the sign function when using infinite terms. In practice, the high-frequency coefficients with relatively lower energy will be ignored to avoid the huge computational overhead, and the sign function will be represented as the

combination of a fixed number of sine functions with different periods. Compared with the existing approximation methods, the proposed frequency domain approximation approach does not affect the low-frequency domain information of the original sign function, *i.e.*, the part that occupies most energy of sign function [50] as shown in Fig. 1. Thus, the main direction of the corresponding gradient *w.r.t.*the original sign function can be more accurately maintained. In addition, to further compensate the subtle approximation error, we explore a noise adaptation module in the training phase to refine the gradient. To verify the effectiveness of the proposed method, we conduct extensive experiments on several benchmark datasets and neural architectures. The results show that our method can surpass other state-of-the-art methods for training binary neural networks with higher performance.

## 2 Approach

In this section, we first provide the preliminaries of BNNs and STE briefly. Then, we propose to estimate sign function with the combination of sine functions and analyze the inaccurate estimation problem. Finally, we introduce a noise adaptation module to solve the problem and derive our FDA-BNN framework.

### 2.1 Preliminaries

The conventional BNNs quantize the weight matrix $W \in \mathbb{R}^{c_i \times c_o \times k \times k}$ and activation matrix $A \in \mathbb{R}^{b \times c_i \times h \times w}$ to 1-bit, and obtain the binary matrices $W_b \in \mathbb{R}^{c_i \times c_o \times k \times k}$ and $A_b \in \mathbb{R}^{b \times c_i \times h \times w}$ for the intermediate convolution layers using sign function as follows:

$$w_b = \begin{cases} 1, & w_f > 0, \\ -1, & w_f \leq 0, \end{cases} \quad a_b = \begin{cases} 1, & a_f > 0, \\ -1, & a_f \leq 0, \end{cases} \tag{1}$$

where $w_f$ and $a_f$ are the elements in the original 32-bit floating point weight matrix $W$ and activation matrix $A$, respectively, and $w_b$ and $a_b$ are the elements in the binarized weight matrix $W_b$ and activation matrix $A_b$, respectively. Given the 1-bit weights and activations, the original convolution operation can be implemented with bitwise XNOR operation and the bit-count operation [30]. During training, both the full-precision weight matrix $W$ and the 1-bit weight matrix $W_b$ are used, while only $W_b$ is kept for inference.

Note that it is impossible to apply back-propagation scheme on the sign function. The gradient of sign function is an impulse function that is almost zero everywhere and is unable to be utilized in training. Thus, the STE method [1] is introduced to compute its gradient as:

$$\frac{\partial \mathcal{L}}{\partial w} = Clip(\frac{\partial \mathcal{L}}{\partial w_b}, -1, 1), \tag{2}$$

in which $\mathcal{L}$ is the corresponding loss function for the current task (*i.e.*, cross-entropy loss for image classification) and

$$Clip(x, -1, 1) = \begin{cases} -1, & \text{if } x < -1, \\ x, & \text{if } -1 \leq x < 1. \\ 1, & \text{otherwise.} \end{cases} \tag{3}$$

Since STE is an inaccurate approximation of the gradient of sign function, many works have focused on replacing STE during back-propagation. BNN+ [6] introduced a SignSwish function, while DSQ [10] proposed a tanh-alike function and minimized the gap between the sign function with a learnable parameter. Such methods focus on approximating sign function in spatial domain, and corrupt the main direction of the gradient. Thus, we propose a new way to estimate sign function to maintain the information in frequency domain.

### 2.2 Decomposing Sign with Fourier Series

In the following sections, we use $f(\cdot)$ and $f'(\cdot)$ to denote an original function and its corresponding gradient function.

Recall that the gradient of sign function is an impulse function which is unable to be back-propagated. Thus, zero-order (derivative-free) algorithms such as evolutionary algorithms need to be applied to reach the optimal solution, which is quite inefficient. Thus, we propose to find a surrogate function

that can be empirically solved by first-order optimization algorithms such as SGD, while theoretically has the same optimal solution as sign function.

It has been proved that any periodical signal with period $T$ can be decomposed into the combination of Fourier Series (FS) [45]:

$$f(t) = \frac{a_0}{2} + \sum_{i=1}^{\infty} [a_i \cos(i\omega t) + b_i \sin(i\omega t)], \tag{4}$$

in which $\omega = 2\pi/T$ is the radian frequency, $a_0/2$ is the direct component and $\{b_i\}_{i=1}^{\infty}$ ($\{a_i\}_{i=1}^{\infty}$) are the coefficients of the sine (cosine) components. Specifically, when the periodical signal is the square wave, we have

$$a_i = 0 \text{ for all } i; \quad b_i = \begin{cases} \dfrac{4}{i\pi}, & \text{if } i \text{ is odd,} \\ 0, & \text{otherwise.} \end{cases} \tag{5}$$

and derive the FS for the square wave $s(t)$:

$$s(t) = \frac{4}{\pi} \sum_{i=0}^{\infty} \frac{\sin((2i+1)\omega t)}{2i+1}. \tag{6}$$

Note that when the signal is restricted into a single period, the sign function is exactly the same as the square wave:

$$sign(t) = s(t), \quad |t| < T. \tag{7}$$

Thus, the sign function can also be decomposed into the combination of sine (cosine) functions, and its derivative is computed as:

$$s'(t) = \frac{4\omega}{\pi} \sum_{i=0}^{\infty} \cos((2i+1)\omega t). \tag{8}$$

In this way, we propose to replace the derivative used in STE (Eq. (2)) with Eq. (8) during back-propagation for better approximating the sign function.

The FS decomposition is a lossless representation of the sign function when using infinite terms by transforming the signal from spatial domain into frequency domain, as shown in Theorem 1. First, we can rewrite Eq. (6) as

$$\hat{s}_n(t) = \frac{4}{\pi} \sum_{i=0}^{n} \frac{\sin((2i+1)\omega t)}{2i+1}, \tag{9}$$

where $n$ is the number of FS terms. The corresponding derivative is

$$\hat{s}'_n(t) = \frac{4\omega}{\pi} \sum_{i=0}^{n} \cos((2i+1)\omega t). \tag{10}$$

In the following, we show that as $n$ increases, the mean squared error between the estimator $\hat{s}_n(t)$ and the ground-truth $s(t)$ decreases, and converges to 0 when $n \to \infty$.

**Theorem 1.** *(Mean Square Convergence [43]) Given $s(\cdot)$ as the square wave function which is integrable in a period, and $\hat{s}_n(\cdot)$ is the $n^{th}$ partial sum of the Fourier Series of $s(\cdot)$, then $\hat{s}_n(\cdot)$ converges to $s(\cdot)$ in the mean square sense,* i.e.,

$$\frac{1}{T} \int_T ||s(t) - \hat{s}_n(t)||_2^2 dt \to 0 \quad as \quad n \to \infty. \tag{11}$$

The above theorem shows that the estimator $\hat{s}_n(t)$ converges to the ground-truth $s(t)$ when using an infinite number of FS terms, thus theoretically has the same optimal solution. In practice, we use a fixed number of FS terms for two reasons: 1) We can avoid the huge computational overhead by ignoring high-frequency coefficients. 2) We can still accurately maintain the main direction of the corresponding gradient *w.r.t.* the original sign function. Compared to other estimation methods that directly estimate sign function in the spatial domain, our method benefits to the approximation process since the proposed approximation approach does not influence the low-frequency domain information of the original sign function, which occupies most of the energy.

## 2.3 Solving the Inaccurate Estimation Problem

Although using infinite FS terms makes the estimator $\hat{s}_n(t)$ converges to the ground-truth $s(t)$, an error

$$r(t) = s(t) - \hat{s}_n(t) \qquad (12)$$

occurs in reality when $n$ is finite. The error comes from the infinite high-order FS terms and has little energy. Basically, a simple way to solve the inaccurate estimation problem is to increase the number of FS terms $n$ to make $\hat{s}_n(t)$ approaches $s(t)$. However, the gradient $\hat{s}'_n(t)$ will approach the impulse function and destabilize the back-propagation process. Besides, increasing the number of FS terms $n$ will increase the training time. Thus, we propose to use another way to solve the aforementioned problem.

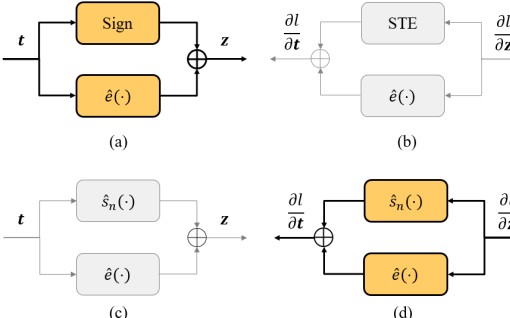

Figure 2: **(a)** The forward pass of the combination of sign function and noise adaptation module. **(b)** Corresponding backward pass of (a). **(c)** The forward pass of the combination of the sine module and the noise adaptation module. **(d)** Corresponding backward pass of (c). The actual forward and backward pass are **bolded** in the figure.

Specifically, we treat the error $r(t)$ between the sign function and the estimation function as noise. In the previous works, there are several ways to deal with noise. The first is to change the training framework to derive a robust network for noise [15, 14, 55, 9]. The second is to improve the training loss function to a noise robust loss [56].

We propose to use a learnable neural network called noise adaptation module to deal with noise. Compared with increasing the number of FS terms, neural network has learnable parameters which make the training more flexible. Besides, the neural network has been proved to be Lipschitz continuous [44, 61] which means the gradient is continuous and avoid the problem from impulse function. We train the noise adaptation module through an end-to-end manner which is reasonable when initialized the weight of the module in a correct way. It is easy to know that the distribution of $r(t)$ in Eq. (12) has zero mean and finite variance regardless of the number of FS terms $n$. Assuming that the input of noise adaptation module is Gaussian distribution, then the initialization of the weights of noise adaptation module need to have zero mean in order to make the distribution of output fit that of $r(t)$. As long as the initial output is unbiased, the proceeding learning process will automatically approach the final goal through an end-to-end training manner with gradient descent based optimization methods.

Specifically, given the input vector $\mathbf{t} \in \mathbb{R}^{1 \times d}$, the process of sign function can be replaced by the combination of a sine module $\hat{s}_n(\cdot)$ and a noise adaptation module $\hat{e}(\cdot)$ which estimates the error $r(\cdot)$ using two fully-connected layers with ReLU activation function and a shortcut:

$$\hat{e}(\mathbf{t}) = \sigma(\mathbf{t}W_1)W_2 + \eta(\mathbf{t}), \qquad (13)$$

in which $\sigma(t) = \max(t, 0)$, $W_1 \in \mathbb{R}^{d \times \frac{d}{k}}$ and $W_2 \in \mathbb{R}^{\frac{d}{k} \times d}$ are the parameters of the two FC layers in which $k$ is used to reduce the number of parameters and is set to 64 in the following experiments. $\eta(\cdot)$ is the shortcut connection.

There are two benefits from adding the shortcut connection. The first is that it helps the flow of gradient and alleviates the gradient vanishing problem [28, 38, 20]. The second is that given $\eta(t) = a \cdot t$ as an example, we have $\eta'(t) = a$ which equals to adding a bias to the gradient. This benefit the training process since given limited training resources and time, only finite number of FS terms $n$ can be used to compute $\hat{s}_n(t)$ in Eq. (9), and raise the problem that the tail of the estimated gradient $\hat{s}'_n(t)$ oscillates around 0 at a very high frequency. This phenomenon does harm to the optimization process, since a very little disturbance on the input will results in two different gradients with opposite direction, and adding a bias term on the gradient helps to alleviate this problem.

Based on the analysis above, given Eq. (9) and Eq. (13), we are able to approximate sign function with the combination of sine module and noise adaptation module:

$$\mathbf{z} = \hat{s}_n(\mathbf{t}) + \alpha \hat{e}(\mathbf{t}), \qquad (14)$$

in which $\alpha$ is the hyper-parameter to control the influence of noise adaptation module. In order to keep the weights and activations as 1-bit in inference, the sine module is used only in the backward pass and replaced as sign function in the forward pass, as shown in Fig. 2. Besides, $\alpha$ is gradually

decreased during training and reaches 0 at the end of training to avoid the inconsistency between training phase and inference phase. The weights of both BNN and noise adaptation module are updated in an end-to-end manner by minimizing the cross-entropy loss $\ell_{ce}$.

---

**Algorithm 1** Feed-Forward and Back-Propagation Process of a convolutional layer in FDA-BNN.

---

**Require:** A convolutional layer with weights $W$ and input features (activations) $A$, the noise adaptation module with parameters $W_{a1}$ ($W_{w1}$) and $W_{a2}$ ($W_{w2}$) for activations (weights), the loss function $\ell_{ce}$, the learning rate $\gamma$, and training iterations $T$.

1: **for** $\tau = 1$ to $T$ **do**
2:     **Feed-Forward:**
3:     Quantize the weights and activations: $W_b = \text{Sign}(W)$, $A_b = \text{Sign}(A)$;
4:     Apply the convolutional operation on the binarized weights and activations: $W_b \otimes A_b$;
5:     **Back-Propagation:**
6:     Compute the gradient of $W_b$ and $A_b$ using standard back-propagation, and get $\frac{\partial \ell_{ce}}{\partial W_b}$ and $\frac{\partial \ell_{ce}}{\partial A_b}$;
7:     Compute the gradient of $A$, and derive $\frac{\partial \ell_{ce}}{\partial A}$ using Eq. (15);
8:     Compute the gradients on parameters of noise adaptation module for activations: $\frac{\partial \ell_{ce}}{\partial W_{a1}}$ and $\frac{\partial \ell_{ce}}{\partial W_{a2}}$ using Eq. (16) and Eq. (17).
9:     Compute the gradients of $W$ and derive $\frac{\partial \ell_{ce}}{\partial W}$, and also $\frac{\partial \ell_{ce}}{\partial W_{w1}}$ and $\frac{\partial \ell_{ce}}{\partial W_{w2}}$, with the same manner in step 7∼8.
10:     **Parameter Update:**
11:     $W \leftarrow \textbf{UpdateParameters}(W, \frac{\partial \ell_{ce}}{\partial W}, \gamma)$;
12:     $W_{a1} \leftarrow \textbf{UpdateParameters}(W_{a1}, \frac{\partial \ell_{ce}}{\partial W_{a1}}, \gamma)$;
13:     $W_{a2} \leftarrow \textbf{UpdateParameters}(W_{a2}, \frac{\partial \ell_{ce}}{\partial W_{a2}}, \gamma)$;
14:     $W_{w1} \leftarrow \textbf{UpdateParameters}(W_{w1}, \frac{\partial \ell_{ce}}{\partial W_{w1}}, \gamma)$;
15:     $W_{w2} \leftarrow \textbf{UpdateParameters}(W_{w2}, \frac{\partial \ell_{ce}}{\partial W_{w2}}, \gamma)$.
16: **end for**

---

Specifically, given Eq. (14) and the loss function $\ell_{ce}$, the gradients in Fig. 2 (d) can be computed as:

$$\frac{\partial \ell_{ce}}{\partial \mathbf{t}} = \frac{\partial \ell_{ce}}{\partial \mathbf{z}} W_2^\top \odot \left((\mathbf{t} W_1) \geq 0\right) W_1^\top$$
$$+ \frac{\partial \ell_{ce}}{\partial \mathbf{z}} \eta'(\mathbf{t})$$
$$+ \frac{\partial \ell_{ce}}{\partial \mathbf{z}} \odot \frac{4\omega}{\pi} \sum_{i=0}^{n} \cos\left((2i+1)\omega \mathbf{t}\right), \quad (15)$$

in which $\frac{\partial \ell_{ce}}{\partial \mathbf{z}}$ is the gradient back-propagated from the upper layers, $\odot$ represents element-wise multiplication, and $\frac{\partial \ell_{ce}}{\partial \mathbf{t}}$ is the partial gradient on $\mathbf{t}$ that back-propagate to the former layer.

$$\frac{\partial \ell_{ce}}{\partial W_1} = \mathbf{t}^\top \frac{\partial \ell_{ce}}{\partial \mathbf{z}} W_2^\top \odot \left((\mathbf{t} W_1) \geq 0\right); \quad (16)$$

$$\frac{\partial \ell_{ce}}{\partial W_2} = \sigma(\mathbf{t} W_1)^\top \frac{\partial \ell_{ce}}{\partial \mathbf{z}}, \quad (17)$$

in which $\frac{\partial \ell_{ce}}{\partial W_1}$ and $\frac{\partial \ell_{ce}}{\partial W_2}$ are gradients used to update the parameters in the noise adaptation module.

The formulations from Eq. (9) to Eq. (17) are applied on both weights and activations for replacing the back-propagation of sign function. The feed-forward and back-propagation process of a convolutional layer in FDA-BNN is summarized in Alg. 1.

# 3 Experiments

In this section, we evaluate the proposed method on several benchmark datasets such as CIFAR-10 [23] and ImageNet [7] on NVIDIA-V100 GPUs to show the superiority of FDA-BNN. All of the proposed FDA-BNN models follow the rule in previous methods that all the convolutional layers except the first and last layers are binarized [57, 3, 6]. All the models are implemented with PyTorch [37] and MindSpore [21].

## 3.1 Experiments on CIFAR-10

The CIFAR-10 [23] dataset contains 50,000 training images and 10,000 test images from 10 different categories. Each image is of size $32 \times 32$ with RGB color channels. Data augmentation methods such as random crop and random flip are used during training. During the experiments, we train the models for 400 epochs with a batch size of 128 and set the initial learning rate as 0.1. The SGD optimizer is used with momentum set of 0.9 and weight decay of 1e-4.

The widely-used ResNet-20 and VGG-Small architectures in BNN literature [30, 10] are used to demonstrate the effectiveness of the proposed method. Dorefa-Net [57] is used as our baseline quantization method, and the sine module and noise adaptation module are used during training. Other competitors include: 1) methods using STE for back-propagation, *e.g.*, BNN [22], Dorefa-Net [57], XNOR-Net [41], TBN [49]; 2) methods using approximated gradient rather than STE during back-propagation, *e.g.*, DSQ [10], BNN+ [6], IR-Net [39]; 3) fine-tuning methods, *e.g.*, LNS [17]. The experimental results are shown in Tab. 1.

Table 1: Experimental results on CIFAR-10 using different 1-bit quantized models. Bit-width (W/A) denotes the bit length of weights and activations, and BackProp denotes the way of computing gradients and updating parameters.

| Network | Method | Bit-width (W/A) | BackProp | Acc (%) |
|---|---|---|---|---|
| ResNet-20 | FP32 | 32/32 | - | 92.10 |
| | Dorefa-Net [57] | 1/1 | STE | 84.44 |
| | XNOR-Net [41] | 1/1 | STE | 85.23 |
| | LNS [17] | 1/1 | STE | 85.78 |
| | TBN [49] | 1/2 | STE | 84.34 |
| | DSQ [10] | 1/1 | Tanh-alike | 84.11 |
| | IR-Net [39] | 1/1 | Tanh-alike | 85.40 |
| | FDA-BNN | 1/1 | Fourier Series | **86.20** |
| VGG-small | FP32 | 32/32 | - | 94.10 |
| | Dorefa-Net [57] | 1/1 | STE | 90.20 |
| | BinaryNet [22] | 1/1 | STE | 89.90 |
| | XNOR-Net [41] | 1/1 | STE | 89.80 |
| | BNN+ [6] | 1/1 | SignSwish | 91.31 |
| | DSQ [10] | 1/1 | Tanh-alike | 91.72 |
| | IR-Net [39] | 1/1 | Tanh-alike | 90.40 |
| | FDA-BNN | 1/1 | Fourier Series | **92.54** |

The results show that the proposed FDA-BNN method outperforms all the competitors and achieves state-of-the-art accuracy. For ResNet-20 architecture, the proposed FDA-BNN achieves an accuracy of 86.20% which improves the baseline model by 1.76%, and is 0.42% higher than the previous SOTA method LNS [17] which is a fine-tuning based method and costs 120 more epochs than our method. When applying the proposed method to VGG-small model, the proposed method achieves an accuracy of 92.54% which is also the highest and improves the baseline model by 2.34%. FDA-BNN method outperforms other gradient estimation methods which approximate sign function in spatial domain such as DSQ [10], BNN+ [6] and IR-Net [39].

## 3.2 Ablation Study

In this section, we conduct several ablation studies to further verify the effectiveness of each component in the proposed method, the impact of shortcut in noise adaptation module and the effect of hyper-parameters.

Firstly, we conduct experiments to understand the effect of sine module and noise adaptation module. The models are verified on CIFAR-10 using ResNet-20 architecture. As shown in Tab. 2, the first line represent the standard Dorefa-Net baseline. We can see that only use sine module can benefit the training process and improve the accuracy from 84.44% to 85.83%. When combining the sine module and noise adaptation module together, we reach the best performance, *i.e.*, 86.20% accuracy, which shows the priority of using both modules in FDA-BNN method during training.

Table 2: Ablation study on sine module and noise adaptation module.

| Sine Module | Noise Adaptation Module | Acc (%) |
|---|---|---|
| × | × | 84.44 |
| ✓ | × | 85.83 |
| ✓ | ✓ | **86.20** |

In order to further verify the usefulness of the noise adaptation module, we plug the module into other gradient approximation methods such as DSQ [10] and BNN+ [6]. The experiments are conducted on CIFAR-10 dataset with ResNet-20 model, and the results in Tab. 3 show that fitting the error between the surrogate function and the original sign function benefits to different gradient approximation methods.

Then we evaluate the influence of using different $\eta(\cdot)$ on the branch of shortcut in noise adaptation module. As shown in Tab. 4, using shortcut results in a better performance than not using it, and the shortcut function $\eta(x) = a\sin(x)$ performs the best during the experiment. Note that this is consistent

Table 3: Plugging noise adaptation module into different methods.

|  | w/o noise module | w/ noise module |
|---|---|---|
| DSQ | 84.11 | **84.46** |
| BNN+ | 84.59 | **84.87** |
| FDA-BNN | 85.83 | **86.20** |

Table 4: Ablation study on $\eta(\cdot)$.

| Functions of shortcut branch | Acc (%) |
|---|---|
| $\eta(x) = 0$ (w/o shortcut) | 85.27 |
| $\eta(x) = ax$ | 85.93 |
| $\eta(x) = a\sin(x)$ | **86.20** |

to the analysis in Sec.2.3. Given $\eta(x) = a\sin(x)$, the gradient flow can still be strengthened. Besides, the gradient $\eta'(x) = a\cos(x)$ equals to adding a positive bias in the given range, and thus can still alleviate the problem that the tail of the estimated gradient $\hat{s}'_n(x)$ oscillates around 0 with a very high frequency. $a$ is set to 0.1 during the experiment.

Using different number of FS terms during training will influence the final performance of BNN. Intuitively, a larger $n$ will reduce the difference between $\hat{s}_n(\cdot)$ and $sign(\cdot)$, but at the same time lead to more time on training since a combination of $n$ gradients will be computed (Eq. (10)). We show the training time spent and the final accuracy achieved by using different $n$ in Fig. 3. We can see that as the number of FS terms $n$ increases, the training time continues to increase while the accuracy saturates at about $n = 10$, which means a moderate $n$ is enough for training an accurate BNN.

As shown in Fig. 3, a small number of FS terms do harm to the performance of BNN while a large $n$ leads to a waste of resources and time. Hence, we explore three different settings of using hyper-parameter $n$ during training.

- Setting 1: the number of FS terms $n$ is fixed to a pre-defined number $n_p$ during training.
- Setting 2: the number of FS terms $n$ is gradually increased from 1 to a pre-defined number $n_p$.
- Setting 3: the number of FS terms $n$ is gradually increased from 9 (best result in setting 1) to a pre-defined number $n_p$.

Results of different settings are shown in Fig. 4. Setting 1 performs well at a wider range than setting 2 and they reach a comparable performance when $n_p$ is large enough. Setting 3 performs the best, since it start from a moderate $n$ and avoid the large gap between $\hat{s}_n(\cdot)$ and $sign(\cdot)$ at the beginning of training, and is able to approach the sign function as the training proceed. Thus, in the following experiments we start $n$ from a moderate number $n_s$ (*e.g.*, $n_s = 9$) and gradually increase it to $n_p = 2n_s$.

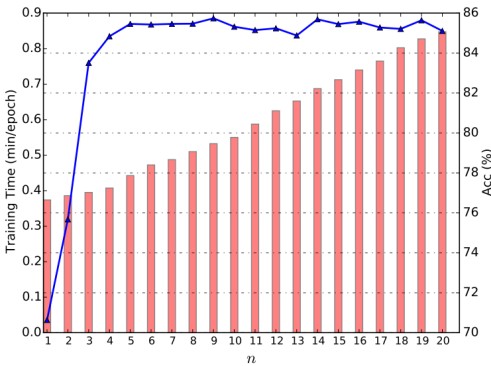

Figure 3: Training time spent and the final accuracy achieved when using different number of FS terms. The blue line denotes the accuracy and the red bar denotes the training time.

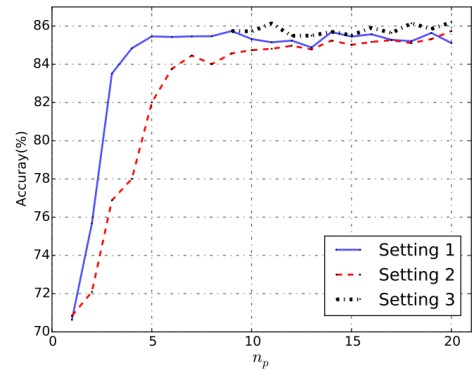

Figure 4: Accuracies under different settings of hyper-parameter $n$ during training.

### 3.3 Experiments on ImageNet

In this section, we conduct experiments on the large-scale ImageNet ILSVRC 2012 [7] dataset. ImageNet dataset contains over 1.2M training images with 224×224 resolutions from 1,000 different

Table 5: Experimental results on ImageNet using different 1-bit quantization methods. 'W/A' denotes the bit-width of weights and activations, and BackProp denotes the way of computing gradients and updating parameters. '*' indicates that ReActNet [29] is used as the baseline method and only the way of computing gradient of sign function is replaced with the proposed method.

| Network | Method | W/A | BackProp | Top-1 | Top-5 |
|---------|--------|-----|----------|-------|-------|
| ResNet-18 | FP32 | 32/32 | - | 69.6 | 89.2 |
| | TBN [49] | 1/2 | STE | 55.6 | 79.0 |
| | Dorefa-Net [57] | 1/1 | STE | 52.5 | 67.7 |
| | Binary-Net [22] | 1/1 | STE | 42.2 | 67.1 |
| | XNOR-Net [41] | 1/1 | STE | 51.2 | 73.2 |
| | ABCNet [26] | 1/1 | STE | 42.7 | 67.6 |
| | Bireal-Net [30] | 1/1 | STE | 56.4 | 79.5 |
| | Bireal-Net [30]+PReLU | 1/1 | STE | 59.0 | 81.3 |
| | BOP [18] | 1/1 | STE | 54.2 | 77.2 |
| | LNS [17] | 1/1 | STE | 59.4 | 81.7 |
| | ReActNet* [29] | 1/1 | STE | 65.5 | 86.1 |
| | HWGQ [4] | 1/2 | Piece-wise function | 59.6 | 82.2 |
| | PCNN ($J = 1$) [11] | 1/1 | DBPP | 57.3 | 80.0 |
| | Quantization Networks [53] | 1/1 | Sigmoid | 53.6 | 75.3 |
| | ResNetE [2] | 1/1 | ApproxSign | 58.1 | 80.6 |
| | BONN [12] | 1/1 | Bayesian learning | 59.3 | 81.6 |
| | IR-Net [39] | 1/1 | Tanh-alike | 58.1 | 80.0 |
| | RBCN [27] | 1/1 | Project-based | 59.5 | 81.6 |
| | RBNN [25] | 1/1 | training-aware | 59.9 | 81.9 |
| | FDA-BNN | 1/1 | Fourier Series | **60.2** | **82.3** |
| | FDA-BNN* | 1/1 | Fourier Series | **66.0** | **86.4** |
| AlexNet | FP32 | 32/32 | - | 56.6 | 80.2 |
| | Dorefa-Net [57] | 1/1 | STE | 43.6 | - |
| | BinaryNet [22] | 1/1 | STE | 41.2 | 65.6 |
| | XNOR-Net [41] | 1/1 | STE | 44.2 | 69.2 |
| | LNS [17] | 1/1 | STE | 44.4 | - |
| | FDA-BNN | 1/1 | Fourier Series | **46.2** | **69.7** |

categories, and 50k validation images. The commonly used data augmentation method is applied during training. The training images are resized, random cropped, random flipped and normalized before training, while the test images are resized, center cropped and normalized.

In the following, we conduct experiments on widely-used ResNet-18 and AlexNet. We use the Adam optimizer with momentum of 0.9 and set the weight decay to 0. The learning rate starts from 1e-3. The experimental results are shown in Tab. 5. Following the acknowledged setting [22, 57], the first and last layers of the network are not binarized for classification, and all methods shown in the table except for ABCNet [26] and Binary-Net [20] do not quantize the down-sample layers in the ResNet shortcut branch.

We can see that on ResNet-18, the proposed FDA-BNN achieves a top-1 accuracy of 60.2% and top-5 accuracy of 82.3%, which improves the baseline method (Bireal-Net + PReLU activation function) by 1.2% and 1.0%, and surpass all other competitors including TBN [49] and HWGQ [4] which use 2-bit activations for quantization. When using ReActNet [29] as the baseline method and replace the way of computing gradient of sign function with our method, the proposed FDA-BNN achieves a top-1 accuracy of 66.0% and top-5 accuracy of 86.4%, which outperforms the baseline method by 0.5% and 0.3%. For AlexNet, we use the quantization method in Dorefa-Net [57] as the baseline method, and achieve a top-1 accuracy of 46.2% and top-5 accuracy of 69.7%, which also outperforms other state-of-the-art methods such as LNS [17] by 1.8% on top-1 accuracy, which shows the superiority of the proposed FDA-BNN method.

# 4   Conclusion

In this paper, we propose a new frequency domain algorithm (*i.e.*, FDA-BNN) to approximate the gradient of the original sign function in the Fourier frequency domain using the combination of a series of sine functions, which does not affect the low-frequency part of the original sign function that occupies most of the energy. Besides, since only finite FS terms are used in reality, a noise adaptation module is applied to fit the error from the infinite high-order FS terms. In order to gradually approach the sign function, we increase the number of FS terms $n$ during the training. The experimental results on CIFAR-10 and ImageNet datasets using various network architectures demonstrate the effectiveness of the proposed method. Binary networks trained using our method achieve the state-of-the-art performance.

**Funding Disclosure**

We thank anonymous area chair and reviewers for their helpful comments. This work was supported in part by the Australian Research Council under Projects DE180101438 and DP210101859.

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
