# OpenReview forum: "Learning Frequency Domain Approximation for Binary Neural Networks"
_NeurIPS.cc/2021/Conference — NeurIPS 2021 Oral_

### Official Review · Reviewer_Xxt6 · 2021-07-13

**Rating:** 8
**Confidence:** 4

**Summary:**

This paper aims to solve the gradient approximation problem in binary neural networks, i.e. replace the non-differentiable sign function with Fourier Series, and then compensate the approximation error with noise adaptation module.

**Limitations And Societal Impact:**

It seems that the initial value of \alpha in Eq.(14) and the way of decreasing it to 0 is not specified in the experiment.

The parameter T in Eq.(7) is also an important parameter. The author should specify the value of T used in the experiment and further explain how to deal with the input value outside T.


**Main Review:**

This paper is technically sound and easy to understand. It is an interesting idea to approximate the gradient in frequency domain and I believe it is enlightening to other BNN gradient approximation articles. Specifically, Fig.1 in the main paper is inspiring and shows that even a nearly perfect approximation function such as tanh can still have an approximation error in the low-frequency domain.

The noise adaptation module is important to compensate the approximation error, and the shortcut in the module basically solves the problem when using Fourier Series that the gradient oscillates around 0 at a very high frequency.

The method does not add any extra parameters or FLOPs, and leave the main architecture unchanged. The strong experimental results on CIFAR-10 and ImageNet shows the usefulness of the proposed method.


**Time Spent Reviewing:**

2

---

> ### Author Response · Authors · 2021-08-04
> **To Reviewer Xxt6**
>
>  Thanks for your constructive comments and support.
>
> Q1: It seems that the initial value of \alpha in Eq.(14) and the way of decreasing it to 0 is not specified in the experiment.
>
> A1: Thanks for your suggestion. The initial value of $\alpha$ is set to 0.1 during the experiment, and a cosine decay is used to decrease it to 0. This will be further specified in the final version.
>
> Q2: The parameter T in Eq.(7) is also an important parameter. The author should specify the value of T used in the experiment and further explain how to deal with the input value outside T.
>
> A2: The input is clipped to [-1, 1] as the other BNN methods do. And then we set the hyper-parameter $\omega$ in Eq.6 to ensure that the range of the clipped input is smaller than T. Note that $\omega=2\pi / T$ as shown in Line 107. Then, all the value of the input will be smaller than T. $\omega$ is set to 0.1 during the experiment. This will also be specified in the final version of the paper.

---

### Official Review · Reviewer_UYYe · 2021-07-14

**Rating:** 8
**Confidence:** 4

**Summary:**

This is a novel gradient estimation method for binary networks based on the Fourier series estimation of sign function. By doing this, the energy in low-frequency domain is preserved, and that of high frequency is estimated by a noise adaptation module.

**Limitations And Societal Impact:**

The noise adaptation module is trained through an end-to-end manner, and there is no specific target used to guide the training, the author should further explain this in the paper.

Although there are obvious improved performances compared to Dorefa net and Bireal net, the author should conduct more experiments on stronger baselines to further demonstrate the usefulness of the proposed method.


**Main Review:**

Basically, the idea of using Fourier series to estimate sign function is straight-forward and easy to understand, but treating it as the frequency domain approximation is interesting. It is a novel gradient estimation method that is orthogonal to a series of binary network algorithms, which means it can fit well into them to further improve their performance.

It is also interesting to solve the inaccurate estimation problem through a learnable neural network, which is the noise adaptation module. While the Fourier series is a surrogate optimizer of sign function, enlarging the FS terms to reduce the estimation error will make it hard to train the binary network. Thus, the author treats the high-order estimation error as noise and use a learnable network to fit the error which makes it easier to train. During training, the weights of the noise module continue to decrease so that the network is finally consistent to that used for inference.

Experimental results on CIFAR-10 and ImageNet show an obvious improved performance compared to the baseline methods.


**Time Spent Reviewing:**

3.5hours.

---

> ### Author Response · Authors · 2021-08-04
> **To Reviewer UYYe**
>
>  Thanks for your constructive comments and support.
>
> Q1: The noise adaptation module is trained through an end-to-end manner, and there is no specific target used to guide the training, the author should further explain this in the paper.
>
> A1: This is a good question. In fact, it is reasonable to train the noise adaptation module through an end-to-end manner. Note that the distribution of the error $r(t)$ in Eq.12 has zero mean and finite variance. And when initializing the weight of noise adaptation module with zero mean, the distribution of output will fit that of $r(t)$. As long as the initial output is unbiased, the proceeding learning process will automatically approach the final goal through and end-to-end training manner with weight decay methods. Also note that this is directly related to the final goal of achieving high accuracy. Compared to the two-step optimization method that first fitting the output of noise module to $r(t)$, and then use the output to optimize the final loss function, the one-step optimization method is always shown to be better since the two-step method is a kind of greedy method that the local optimum in each step may not converge to the global optimum for the final goal.
>
> Q2: Although there are obvious improved performances compared to Dorefa net and Bireal net, the author should conduct more experiments on stronger baselines to further demonstrate the usefulness of the proposed method.
>
> A2: We use the same experimental setting as ReactNet [2] do, and only change the way of computing gradient of sign function with our proposed method. The experimental results of ResNet-18 on ImageNet dataset are shown below:
>
> |model| Top1 acc(%)| Top5 acc(%)|
> |-|-|-|
> |ReactNet| 65.5| 86.1|
> |Real2Bin| 65.4| 86.2|
> |ours| 66.0| 86.4|
>
> The results show that we improve the ReactNet top-1 performance by 0.5% and top-5 performance by 0.3%, and improve the Real2Bin top-1 performance by 0.6% and top-5 performance by 0.2%. The experimental results will be added in the final version of the paper.
>
> [1] Training binary neural networks with real-to-binary convolutions.
>
> [2] Reactnet: Towards precise binary neural network with generalized activation functions.

---

### Official Review · Reviewer_oG66 · 2021-07-14

**Rating:** 8
**Confidence:** 4

**Summary:**

BNN is an important network when we cannot support huge computational costs. However, its optimization is still an open question and cannot be addressed well. This paper proposed a novel gradient computation method for sign operation in BNN. The author used the combination of sine functions to approximate sign operation, and a noise adaptation module is applied to fit the high order error between sine functions and sign operations.


**Limitations And Societal Impact:**

Relatively weak baselines such as Dorefa-Net and Bireal-Net+PReLU are used. I understand the rebuttal time is limited, thus it is optional but better that the author conduct experiments based on a stronger baseline such as ReactNet [1].

[1] ReActNet: Towards Precise Binary Neural Network with Generalized Activation Functions.

**Main Review:**

Pros:

1. Basically, this is a simple, novel, and effective method that approximates sign operation in the frequency domain aspect, which is different from other gradient approximation methods. As all known, previous studies only consider the optimization in the spatial space. This paper revisits this problem in the frequency domain, which is very interesting to me.

2. The paper is technically sound and adding a noise adaptation module is a clever way to fit the high order error. The experimental results demonstrate the effectiveness of the proposed training method.

3. In experiments, both CIFAR10 (10 classes) and ImageNet (1000 classes) are used to validate the proposed method. The results show that the proposed method is accurate and efficient.

Cons:

1. Figure 1 is important to help readers understand why your motivation is working. More explanation should be added in the caption.

2. Notations are unclear. In Section 2.1, there are some notations that are not explained, e.g., c_i, h…

3. Some parts are not self-contained. In Theorem 1, what is the convergence rate regarding n?

--------------------------------
POST REBUTTAL
--------------------------------

After reading comments from other reviewers, I am confident that this paper should be accepted by NeurIPS 2021. Thus, I would like to increase my score to 8.

**Time Spent Reviewing:**

6hrs

---

> ### Author Response · Authors · 2021-08-04
> **To Reviewer oG66**
>
>  Thanks for your constructive comments and support.
>
> Q1: Figure 1 is important to help readers understand why your motivation is working. More explanation should be added in the caption.
>
> A1: Thanks for your suggestion. For different columns, we show different properties of sign function, proposed Fourier Series, Tanh function and SignSwish function. For different rows, the original function, the function in frequency domain and the difference between current function and the sign function in frequency domain of the specific functions are shown. The caption of Fig.1 will be thoroughly explained in the final version.
>
> Q2: Notations are unclear. In Section 2.1, there are some notations that are not explained, e.g., c_i, h…
>
> A2: $c_i$ and $c_o$ represents the number of input and output channels and $h$ and $w$ is the height and width of the feature. This will be further explained in the final version of the paper.
>
> Q3: Some parts are not self-contained. In Theorem 1, what is the convergence rate regarding n?
>
> A3: It has been shown that Fourier Series converges at a rate of $\mathcal O(n^{-2})$ in [R1]. This will be added in the final version for integrity.
>
> Q4: Relatively weak baselines such as Dorefa-Net and Bireal-Net+PReLU are used. I understand the rebuttal time is limited, thus it is optional but better that the author conduct experiments based on a stronger baseline such as ReactNet [1].
>
> A4: We use the same experimental setting as ReactNet [1] do, and only change the way of computing gradient of sign function with our proposed method. The experimental results of ResNet-18 on ImageNet dataset are shown below:
>
> |model| Top1 acc(%)| Top5 acc(%)|
> |-|-|-|
> |ReactNet| 65.5| 86.1|
> |ours| 66.0| 86.4|
>
> The results show that we improve the ReactNet top-1 performance by 0.5% and top-5 performance by 0.3%. The experimental results will be added in the final version of the paper.
>
> [R1]  Iserles, A., Nørsett, S.P., From high oscillation to rapid approximation I: Modified Fourier expansions.

---

### Official Review · Reviewer_f6JJ · 2021-07-15

**Rating:** 7
**Confidence:** 4

**Summary:**

This work focuses on training binary neural networks. Instead of using the traditional STE method to estimate the gradient of sign function whose gradient is almost everywhere which cannot be used for back-propagation, the paper proposes to estimate it with Fourier series in the Fourier frequency domain. To avoid using infinite terms of Fourier series, an additional noise adaptation module is further applied during training.


**Limitations And Societal Impact:**

-	In table 2, it seems that the ablation study is not complete. I also want to know what is the final accuracy when only use the noise adaptation module during training. I understand that the performance may not be good when using noise adaptation module along, but the result should still be given for integration.

-	In table 4, it seems that the shortcut is important during training. The author should further explain this.

-	What is the number of FS terms used when conducting experiments on ImageNet? Is it the same with that used for CIFAR-10?

-	The author does not compare to the state-of-the-art BNN methods such as real2bin [1] and reactnet [2].

[1] Training binary neural networks with real-to-binary convolutions.

[2] Reactnet: Towards precise binary neural network with generalized activation functions.


**Main Review:**

+ Compared to the previous gradient estimation methods such as using tanh function, signswish function, etc, the proposed method does not contaminate the low frequency part of the sign function in the frequency domain perspective which occupies most of the energy, thus keep the main direction of the factual gradient.

+ Estimating sign with Fourier series is reasonable, and a theorem is provided to show the rationality.

+ The high frequency part of Fourier series is ignored to avoid the huge computational cost. Instead, a noise adaptation module is used to solve the inaccurate estimation problem. The weight of the noise adaptation module is gradually decreased during training. This is a good compromise between finding the correct direction of the gradient and the consistency between training and inference process.

+ There are a bunch of ablation studies to verify the effectiveness of sign module and noise adaptation module, and the hyper-parameter used during the experiments.

+ The experiments on CIFAR10 and Imagenet show that the proposed method can benefit from the Fouries series estimation.


**Time Spent Reviewing:**

3 hours

---

> ### Author Response · Authors · 2021-08-04
> **To Reviewer f6JJ**
>
>  Thanks for your constructive comments and support.
>
> Q1: In table 2, it seems that the ablation study is not complete. I also want to know what is the final accuracy when only use the noise adaptation module during training. I understand that the performance may not be good when using noise adaptation module alone, but the result should still be given for integration.
>
> A1: The noise adaptation module is used to estimate the 'residual error' between the sign function and the approximation function. Thus, it will yield a relatively poor results when using alone. We conduct experiments with ResNet-20 on CIFAR-10 dataset by using only Noise Adaptation module and the final accuracy is $77.16%$. The experimental result shows that it is necessary to use sine module to estimate sign function, and combining it with noise adaptation module will yield a better result.
>
> Q2: In table 4, it seems that the shortcut is important during training. The author should further explain this.
>
> A2: We give the analysis of shortcut connection in Line 175-182. In fact, directly using sine module to estimate sign function will result in a problem that the tail of the estimated gradient oscillate around 0 at a very high frequency, which does harm to the optimization process since a very little disturbance on the input will results in two different gradients with opposite direction. Thus, adding a shortcut connection equals to adding a bias to the gradient which alleviate this problem. Besides, shortcut connection helps the gradient flow and avoid the gradient vanishing problem.
>
> Q3: What is the number of FS terms used when conducting experiments on ImageNet? Is it the same with that used for CIFAR-10?
>
> A3: We use the same setting as in CIFAR-10 (as shown in Line 276-277) but a different start number $n_s$. When conducting experiments on ImageNet, we use $n_s=6$. This will be added in the final version of the paper.
>
> Q4: The author does not compare to the state-of-the-art BNN methods such as real2bin [1] and reactnet [2].
>
> A4: We use the same experimental setting as ReactNet [2] do, and only change the way of computing gradient of sign function with our proposed method. The experimental results of ResNet-18 on ImageNet dataset are shown below:
>
> |model| Top1 acc(%)| Top5 acc(%)|
> |-|-|-|
> |ReactNet| 65.5| 86.1|
> |Real2Bin| 65.4| 86.2|
> |ours| 66.0| 86.4|
>
> The results show that we improve the ReactNet top-1 performance by 0.5% and top-5 performance by 0.3%, and improve the Real2Bin top-1 performance by 0.6% and top-5 performance by 0.2%. The experimental results will be added in the final version of the paper.

---

> ### Comment · Reviewer_f6JJ · 2021-08-31
> **Raise my rating score**
>
>
> The authors have solved my concerns greatly.
>
> And I would like to raise my score and vote for acceptance.
>
> Best,
>
> Reviewer f6JJ

---

### Decision · Program_Chairs · 2021-09-28

**Decision:**

Accept (Oral)

**Comment:**

The paper worked on binarized neural networks and proposed to find better surrogates of the sign function in the frequency domain instead of the spatial domain. The idea is motivated, novel, and simple --- I am quite surprised such a simple idea can work fairly well, as a simple idea that works is the greatest to me. The motivation and the key to its success is that approximating the sign function in the frequency domain would drop some high-frequency terms, which cannot affect the quality of gradients much since the gradient directions are mainly determined by the low-frequency terms. The proposed noise adaptation module puts some additional technical difficulty/quality into the paper and is what I am mainly interested in (I have gone through the paper by myself and asked some questions in a separate post).

Although there were some concerns in the beginning, the authors have done a particularly good job in their rebuttal. In the end, all the reviewers hold very positive opinions and vote for acceptance. I think the idea (i.e., frequency domain approximation plus noise adaptation layer) can also be used in other areas besides BNN, since non-smooth functions are common and straight-through estimator is popular in deep learning or the entire machine learning. I am sure that this paper will be impactful and therefore I recommend it as an oral presentation.

**Consistency Experiment:**

NeurIPS has a long history of experimentation. In 2014, NeurIPS ran an experiment in which 10% of submissions were reviewed by two independent committees to quantify the randomness in the review process. This year, we repeated a variant of this experiment to see how the quality of the review process has changed over time.  This paper was part of the experiment and was therefore assigned to two committees (consisting of reviewers, an Area Chair, and a Senior Area Chair) that reached independent decisions.  If both committees made the same recommendation, this recommendation was followed. If a single committee recommended acceptance, the paper was accepted (with the exception of a few cases in which the other committee identified what we considered a fatal flaw, e.g., an error in a key result).

This copy’s committee reached the following decision: **Accept (Oral)**

The other committee assigned to the paper recommended **Accept (Poster)**.  You can find the other set of reviews, along with any follow up discussion with the authors here:
https://openreview.net/forum?id=kwN2xvZ2XZ9